# Characterization of Municipal Solid Waste and Assessment of Its Potential for Refuse-Derived Fuel (RDF) Valorization

Khadija Sarquah [1,2,*], Satyanarayana Narra [1,3], Gesa Beck [2], Uduak Bassey [1,2], Edward Antwi [1], Michael Hartmann [2], Nana Sarfo Agyemang Derkyi [4], Edward A. Awafo [4] and Michael Nelles [1,3]

[1] Department of Waste and Resource Management, University of Rostock, 18051 Rostock, Germany
[2] Berlin School of Technology, SRH Berlin University of Applied Sciences, 10587 Berlin, Germany
[3] German Biomass Research Centre (DBFZ), 04347 Leipzig, Germany
[4] School of Engineering, University of Energy and Natural Resources, Sunyani P.O. Box 214, Ghana
[*] Correspondence: khadija.sarquah@uni-rostock.de

**Abstract:** Reuse and recycling are preferred strategies in waste management to ensure the high position of waste resources in the waste management hierarchy. However, challenges are still pronounced in many developing countries, where disposal as a final solution is prevalent, particularly for municipal solid waste. On the other hand, refuse-derived fuel as a means of energy recovery provides a sustainable option for managing mixed, contaminated and residual municipal solid waste (MSW). This study provides one of the earliest assessments of refuse-derived fuel (RDF) from MSW in Ghana through a case study in the cities of Accra and Kumasi. The residual/reject fractions (RFs) of MSW material recovery were characterized for thermochemical energy purposes. The studied materials had the potential to be used as RDF. The combustible portions from the residual fractions formed good alternative fuel, RDF, under the class I, II-III classification of the EN 15359:2011 standards. The RDF from only combustible mixed materials such as plastics, paper and wood recorded a significant increase in the lower heating value (28.66–30.24 MJ/kg) to the mass RF, with the presence of organics (19.73 to 23.75 MJ/kg). The chlorine and heavy metal content met the limits set by various standards. An annual RDF production of 12 to 57 kilotons is possible from the two cities. This can offset 10–30% of the present industrial coal consumption, to about 180 kiloton/yr $CO_2$ eq emissions and a net cost saving of USD 8.7 million per year. The market for RDF as an industrial alternative fuel is developing in Ghana and similar jurisdictions in this context. Therefore, this study provides insights into the potential for RDF in integrated waste management system implementation for socioeconomic and environmental benefits. This supports efforts towards achieving the Sustainable Development Goals (SDGs) and a circular economy.

**Keywords:** refuse-derived fuel (RDF); energy recovery; municipal solid waste; co-combustion; alternative fuel; thermochemical valorization

## 1. Introduction

Municipal solid waste (MSW) represents a valuable resource for energy recovery in the form of heat and electricity, syngas, char and pyrolysis oils. These strategies exist to reduce the quantity of municipal solid waste landfilled without energy recovery and improve alternative materials and energy use [1]. One viable option is alternative fuel concepts such as RDF from MSW utilized in cement kilns, boilers and for power generation. RDF is produced from mixed combustible portions of waste, including household, commercial and industrial/trade waste [2]. RDF from MSW comprises combustibles such as plastics, paper materials and wood. Research shows that RDF has improved properties such as heating value compared to mass MSW considering product [3] and thermal processes [4,5]. Currently, RDF is largely produced and utilized in countries such as Germany, Italy, the US and the UK for fossil fuel substitution. A growing interest in utilizing RDF is also reported from developing countries such as India, Namibia and Mozambique [6]. In Europe,

12 million tons of RDF replaced fossil fuel in the cement and waste-to-energy (WtE) plants in 2015 [7]. According to reports [8], RDF utilization has reached between 30% to 60% of fossil fuel and alternative material substitution in Europe. Other energy carriers such as fuel oil and gases are obtained from RDF through pyrolysis and gasification processes [6].

There are several developing countries such as Ghana where MSW management continues to be a major environmental challenge, affecting sanitation, land use, water and natural resources [9]. Growing urban populations, increased MSW generation per capita, insufficient financing, legislation gaps, lack of strategic planning and impacts of uncontrolled landfills are significant challenges in the waste sector [10]. MSW generation in Ghana is estimated at around 14.5 kilo tons/day. The common practice of mixed MSW also decreases the recycling potential as a result of contamination [11], hence the low rates recorded [12]. The waste sector in Ghana was ranked third among sources of GHG emissions (3.2 $MtCO_2eq$), mainly methane and $CO_2$ from waste disposal and fuel use, while industrial processes accounted for 1.52 $MtCO_2eq$ according to the 2019 inventory report [13]. Similarly, the most energy consumed in Ghana derives from fossil fuels, followed by biomass in the form of firewood and charcoal and electricity, representing 50%, 34% and 17% respectively in 2021 [14]. In addition, several industrial processes are powered by fossil fuels: coal, diesel and electricity from natural gas. None is reported to utilize any form of alternative fuels from MSW, coupled with the high cost of current fuel sources increasing the operational cost for industries [15]. Currently, about nine different cement companies are in operation in Ghana [16], with a few operating their kilns using fossil fuel sources of energy [17]. This suggests the potential for dependable alternative fuel and other suitable raw material substitution in the sector.

The potential for waste-to-energy (WtE) from MSW has been assessed within the Ghanaian context over the years. However, reports argue that several studies undertaken on WtE favor biochemical processes due to the high portion of organics such as food waste in MSW with little emphasis on thermochemical treatments [18]. Furthermore, Barnor et al. [19] reported that nearly 700 GWh of biogas is possible annually from municipal solid waste with a plant availability of 97%. Thus, about 1.5% of Ghana's total electricity demand represents savings of USD 5 million from MSW to electricity. The study recommended mechanical biological treatment (MBT) primarily because it can separate the organic component of MSW from the inorganic component with the possibility of obtaining non-biodegradable resources for other purposes. Using the MSW decision support tool, Brown et al. [20] assessed the performance of municipal solid waste disposal operations in one of Ghana's municipalities, Wa. The study recommended an integrated system consisting of separation, composting, incineration and RDF to landfill disposal within the municipality since it had a lesser impact on health. However, the study did not qualitatively or quantitatively assess the RDF potential. Afrane et al. [21] also highlighted the dearth of data on the waste management industry in Ghana, which is also a limitation, especially for WtE.

In recent times, the presence of material recovery and recycling facilities has contributed to treating portions of MSW for compost and recycling. On the other hand, non-recyclables of combustible fractions end up at disposal sites as a result of contamination from mixed MSW, compromised physical state and low market value fractions, etc. Thus, there is room for considering combustible portions for thermochemical treatments via the RDF technique for alternative fuel. The availability of resources (MSW) and end-users such as cement kilns suggests that the production and use of RDF is viable. This may also improve on solid waste management in Ghana without major investment into new systems. The use of residual fractions involves separating noncombustible fractions, resizing in some cases, reducing moisture and forming a homogenous product suitable for specific needs [22]. In this context, researchers [3,23,24] have analyzed activities of various MBT plants and landfills to assess the usefulness of MSW residual fractions for RDF. Combustible materials of high calorific values (>20 MJ/kg) and compliance with standards were assessed. It is clear from the previous studies the characteristics of RDF vary considerably

with location (regional and local variations), hence its implications. Furthermore, other variables of relevance, such as drivers and barriers, are imperative for decision-making recommendations. Currently, RDF from MSW and its utilization are relatively new concepts and are largely untapped in Ghana. RDF production as part of an integrated waste management process also falls in line with the waste management hierarchy.

In this context, the aim of this study was to investigate the potential utilization of MSW reject/residual fractions in RDF, the characteristics of RDF and the possible utilization of RDF as a substitute fuel. The study closes a knowledge gap with one of the first case studies to technically evaluate RDF as an improvement of current systems. As a result, this work serves as a reference for the various stakeholders in harnessing the energy potential of RDF from MSW.

## 2. Materials and Methods

### 2.1. Description of Study Locations

Ghana is situated in West Africa, with a total area of 238,533 sq. km, a 550 km coastal line and a tropical climate and a population of about 30.8 million people [25]. Ghana is divided into 16 administrative regions shown in Figure 1, which are further divided into metropolitan, municipal and district assemblies (MMDAs). The study was carried out in three established waste management (material recovery and compost) facilities in the Accra and Kumasi municipalities. The Greater Accra region is the administrative capital of Ghana, a coastal region and the smallest in the country but the most populated among the 16, while Kumasi is the most urban center within the forest regions.

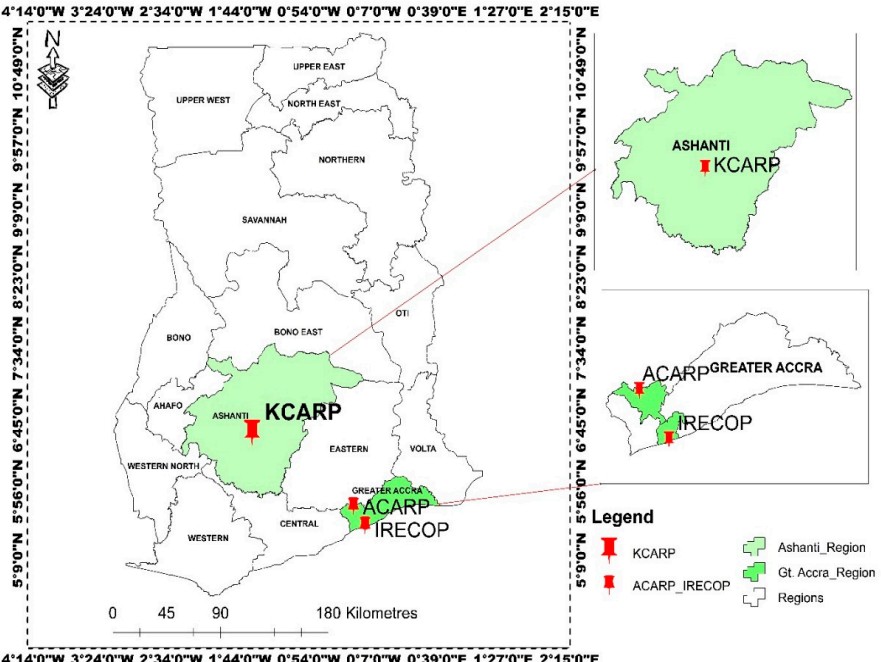

**Figure 1.** Map of Ghana showing the study area (created by the authors).

The facilities are represented in Figure 1, IRECOP, ACARP and KCARP (F1, F2, F3, respectively). They receive MSW and sort into compostable (organic fractions), recyclables (plastics, cartons/paper, metals, etc.) and others (to landfill). The MSW received at these facilities comprises waste from households and commercial areas, but no hospital, hazardous or agricultural waste, with a capacity of about 200, 300 and 1200 tons/day, respectively. Facilities 1 and 2 are located in Accra, while facility 3 is located in Kumasi. The basic flow of process features at these facilities in terms of operation is shown in Figure 2. The MSW received at the site goes through a manual and mechanical sorting process to recover organic fractions for compost and recyclables. Bulky materials, metal scraps, cartons and

glass are removed, and bags are opened to avoid blockages. The organic fractions are sorted mechanically from a trommel (<60 mm) and processed into compost by biological treatment. Fractions after the trommel stage (>60–80 mm) are carried over to the next stage for manual selective sorting where plastics, cans and paper are recovered. Through magnetic separators, ferrous metals are also sorted. The material remaining as the residual/reject fraction (RF), is transferred to the adjoining landfill sites. The RF comprises all other MSW materials that were not recovered in the stages.

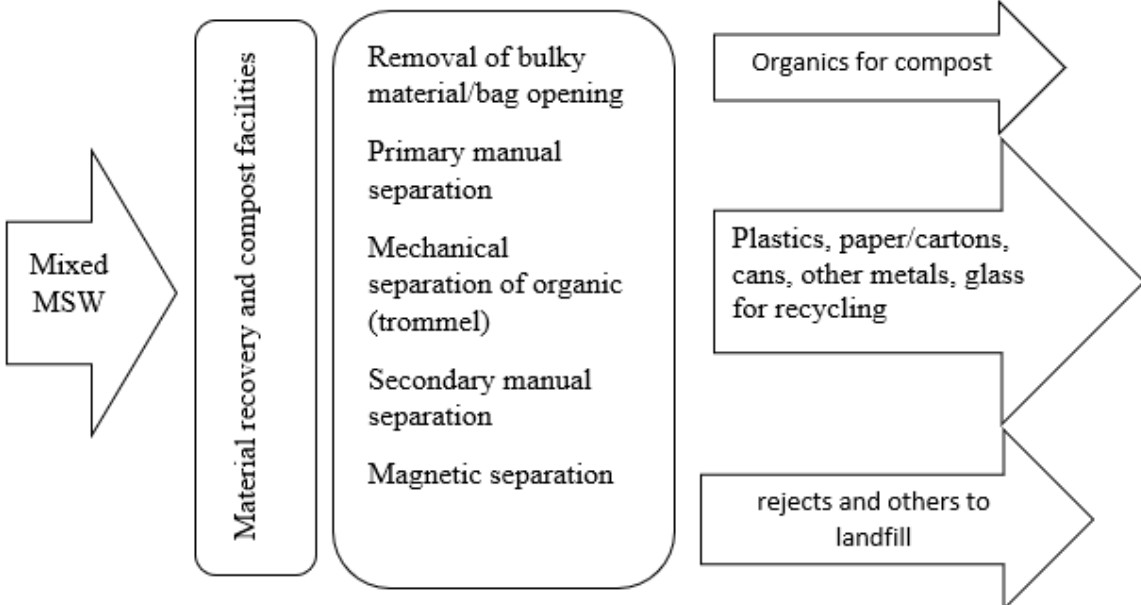

**Figure 2.** The general flow of the sorting process at the MRFs.

*2.2. Characterization*

To evaluate the feasibility of RDF from the waste resources present, composition and characteristics were assessed. The characterization was carried out to assess the composition of both for the received waste and the RF from the sorting process going to the landfill. Sampling campaigns were carried out a week consecutively in the facilities from May to July 2021, following the quartering methods for waste composition [26] and methods in other studies [23,27] to obtain a representative sample. At each sampling, a load was randomly selected, discharged in an area away from other operations and divided into 4 parts, and then two opposite parts were selected and re-divided into four other parts until a sample of approximately 100–120 kg was obtained for sorting. There were no previous data on the RF from the operations in the facilities. The sampled waste was manually sorted into various fractions of organics (food waste, yard waste), plastics, mixed paper (office paper, books, cartons, newsprint), textiles, leather, metals, glass, wood, inert and other waste and weighed in situ. Plastic types were also further differentiated with their plastic identification numbers (PINs) and a MicroNIR spectrometer 1700ES from Analyticon Instruments GmbH [28] and separated. Mass percentages of the fractions obtained were determined, dried and stored for further analysis.

The further analysis aimed to determine the specific properties classifying the waste material as RDF. These comprised the dry matter, ash content, calorific value and elemental analysis, including heavy metal and trace element determination. The samples were put in a preheated oven at 105 °C for 24 h for drying. The dry matter was determined from daily sampling and averages recorded. Dry matter for each was then obtained by

Moisture Content (M) % = (weight of dried sample/Weight of wet sample) * 100%.　(1)

The ash content was determined following EN 15403. The samples were combusted in a muffle oven. The ash content was obtained by calculating the percentage of the residual mass after the combustion as in Equation (2). The experiment was performed in triplicate and the average taken. The ash content was calculated by

$$\text{Ash Content (\%)} = [(\text{sample weight before combustion-sample weight after combustion})/\text{sample weight before combustion}] * 100\%. \tag{2}$$

The elemental composition (CHN) was determined using a Vario Macro cube elemental analyzer, following EN 15407:2011. This was of interest as C content determines the combustibility of the fuel, which also informs its heating value and $CO_2$ as a combustion product. The N and S are essential for assessment especially when applying thermal treatments to RDF, thus predicting the NOx and $SO_2$ formation. Cl content results in HCl as a combustion product which has corrosion and slag potential at a higher content [29]. The Cl and S were determined by the method EN 15408:2011 using ion chromatography (Metrohm 883 Basic plus). Heavy metal and trace element concentrations were determined by following EN 15411:2011 using inductively coupled plasma mass spectrometry (ICP-OES). This composition of the ash also helps to analyze the ash melting, deposit formation and slagging potential. All experiments were performed in triplicate, and the average was determined.

The calorific value represents the energy content. The gross calorific value was determined using an IKA C 6000 bomb calorimeter following EN 15400:2011/EN 51900-1 and 51900-2, while the net was determined by Equation (3). The average characteristics of the RDF obtained were used to determine the classes of the RDF according to EN15359:2011.

$$\text{LHV} = [\text{HHV} - 206\,\text{H}] * [(1 - 0.01\text{MC}) - 23.0\text{MC}] \tag{3}$$

### 2.3. Refuse-Derived Fuel Potential from the Residual Fractions

Based on the findings from the characterization and data from other studies, the quantity and potential use were evaluated. The methodology followed methods from the previous studies [30–32]. RDF quantities were estimated based on the quantities of MSW at the MRFs and mass percentages of the residual fractions. An estimation of the $CO_2$ emissions and cost savings by replacing coal combustion with RDF was included. This is attainable with fuel from waste materials that carry heat values equal to or greater than 20 MJ/kg [33]. Other parameters included the average annual industrial coal consumption of approx. 166,000 tons/yr [34], the calorific value of coal consumed and the cost of coal in Ghana [35]. Studies also report the RDF to coal replacement ratio of 1.42:1 [32,36]. The $CO_2$ emission from coal combustion is 2.42 tons of $CO_2$ per ton of coal [37]. The estimated cost of producing 1 ton of RDF is USD 24.24 [38]. The theoretical estimations were performed from the following equations.

$$\text{RDF generation (ton/yr)} = \text{MSW (ton/yr)} * \text{Residual fraction (wt\%)} * \text{RDF components(wt \%)} \tag{4}$$

$$\text{Coal savings (ton/yr)} = \text{substitution rate} * \text{RDF quantity (ton/yr)} \tag{5}$$

$$\text{Cost of coal saving (USD/yr)} = \text{coal saving (ton/yr)} * \text{cost of coal (USD/ton)} \tag{6}$$

$$\text{Coal combustion } CO_2 \text{ emissions saving (ton/yr)} = \text{coal saving (ton/yr)} * \text{emission factor} \tag{7}$$

## 3. Results and Discussion

### 3.1. Characterization

In Figure 3, the average composition of MSW received at the three facilities in weight percentage is shown; the waste was segregated into the components outlined, and the plastic fractions were further separated. The received MSW at the three facilities comprises on average 48.6% organics, 7.1% paper, 20% plastics, 4.2% textiles, 1.2% metals, 1.0% glass, 0.3% wood, 4.8% sanitary waste, 1.5% leather, 0.6% e-waste, 5.6% inert (sand, ash,

stones) and 5.3% other waste. The composition of the waste obtained supports studies on MSW characteristics in most developing countries, especially Africa [39] in terms of the high organic fraction (40–60%) proportion of the whole waste. This is unlike the composition of MSW from developed economies such as in the EU, with high dry fractions (>50%) and about 25% kitchen waste [40]. However, the organic fraction recorded was lower (average < 48%) than those reported from household waste characterization studies (50–61%) [41]. The composition obtained was comparable to studies from landfill or disposed waste studies conducted in Ghana [42–44]. It can be concluded that waste received at MRFs has similar a composition to that when it is disposed. The composition of the residual/reject fraction (RF) at each facility is presented in Figure 4. This is referred to as the coarse fraction (>60–80 mm sizes) in this study. On average, it comprises 24% organics, 9.2% paper, 34.7% plastics, 9.7% textiles, 1.3% metals, 0.4% glass, 1.2% wood, 6.7% sanitary waste, 2.2% leather, 0.8% e-waste, 4.4% others and 4.8 inert waste (sand, ash, ceramics).

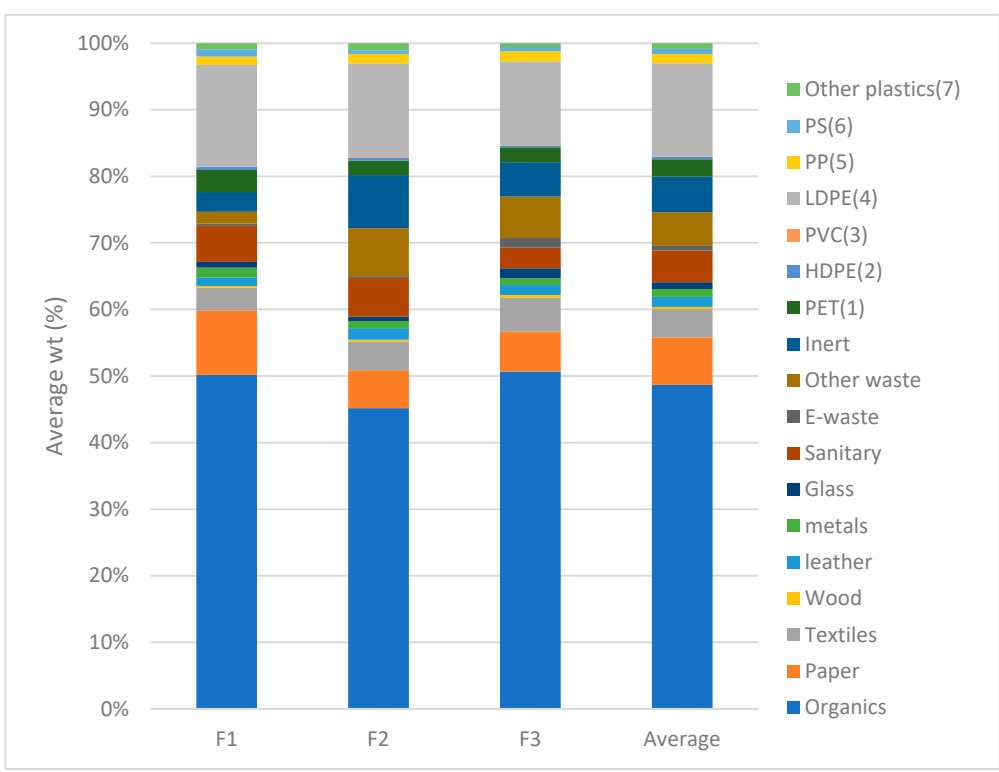

**Figure 3.** Composition of MSW received at the MRFs.

The differences in RF compositions result from the sorting system efficiency and composition of the waste. On average, plastics make up the largest fraction of the RF at around 35%, and LDPE forms the largest plastic type (76%). Generally, high LDPE was found both in the received waste and the residual plastics. This is because LDPE is the widely used plastic type in shopping bags and food packaging. As PVC may increase Cl content, low or no content is required in fuel for thermochemical applications [45]. PVC was rarely noticed in the waste collected (<0.1%). Low fractions were also recorded for HDPE (<3%) and PET (<5%) in the mainstream waste. It was observed that HDPE and PET particularly are collected separately in most urban areas, hence least from the collected waste.

Organic fractions, usable for compost, form about 24.0% of the RF. The quantity could be reduced by further sorting or improving waste bag splitting and opening techniques. Fractions of metals in the RF could also be reduced in the RF by improving the electromagnetic system and introducing eddy current techniques for nonferrous metals. Some hazardous materials such as batteries and electronics were also present in the RF, which is

unsuitable for compost and combustibles. Consequently, the practice of source separation can decrease the level of such contamination as observed in other studies [46,47], especially from the organics.

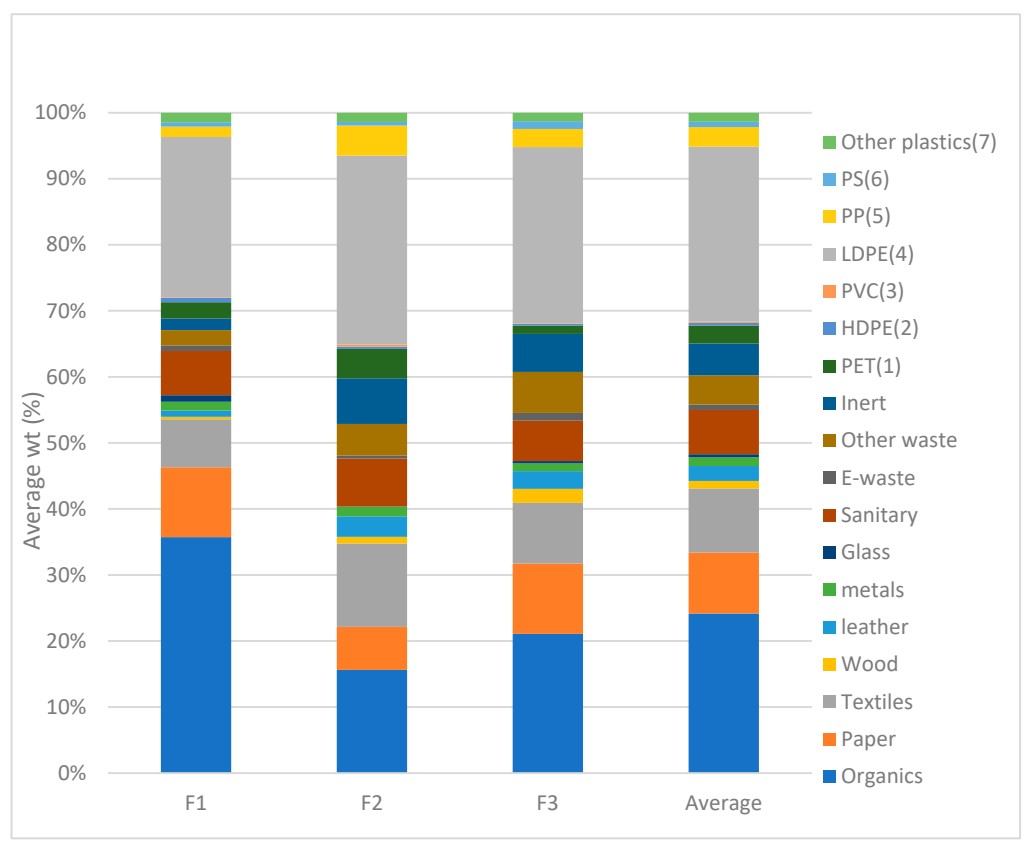

**Figure 4.** Composition of residual fractions (RFs) from the MRFs.

### 3.2. Refuse-Derived Fuel from Residual Fractions

About 70% of the residual fractions in Figure 4 are combustible fractions (including plastics, paper, textiles, leather, wood and others) for possible RDF production. After the waste was sorted into various components, each component's mass and weight percentages were determined. From the average compositions, two RDF types were considered from each facility sampling, with and without organics, and the average composition is presented in Figure 5. Type 1 (T1, T2, T3) represents only combustible fractions from the RF, while type 2 (T4, T5, T6) represents all the RFs for RDF (except for the non-combustible fractions such as metals and glass). The average RDF from this study has more plastic content than in studies from the other context, in which the paper component recorded the highest fraction (Figure 6). Similarly, RDF originating from other developing countries also recorded high plastic relative to other components. This is potentially attributed to the high level of single-use plastics and contamination levels in plastics for other uses in such context [48,49]. From the specific weight of components, the RF was on average about 30–48% by weight of the initial waste, compared with 43.1% [24], 33.4% [23] and 42% [3] recorded from other studies. Hence, the suitability of the RF for RDF production is enshrined in achieving sustainability within the practices of waste management.

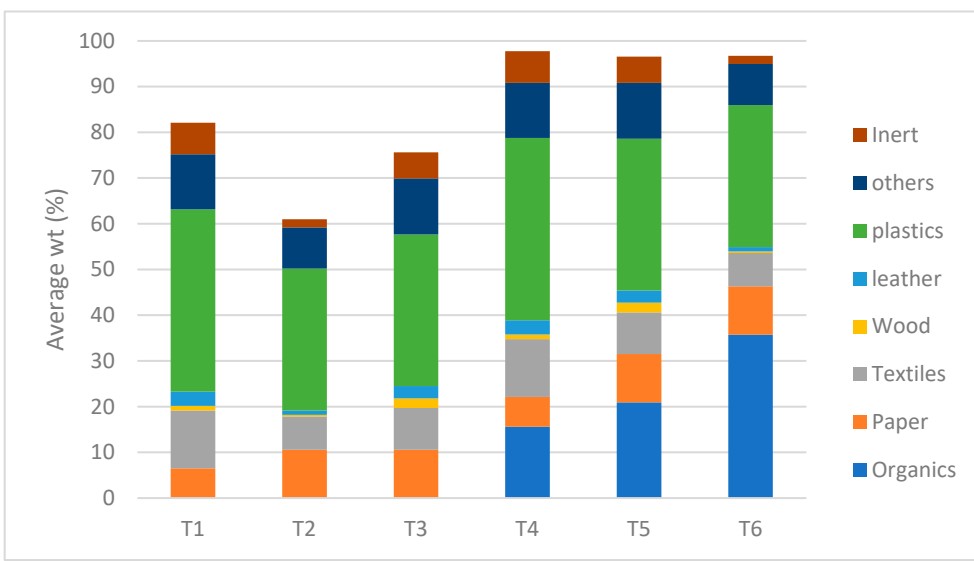

**Figure 5.** The average composition of RDF from the RF.

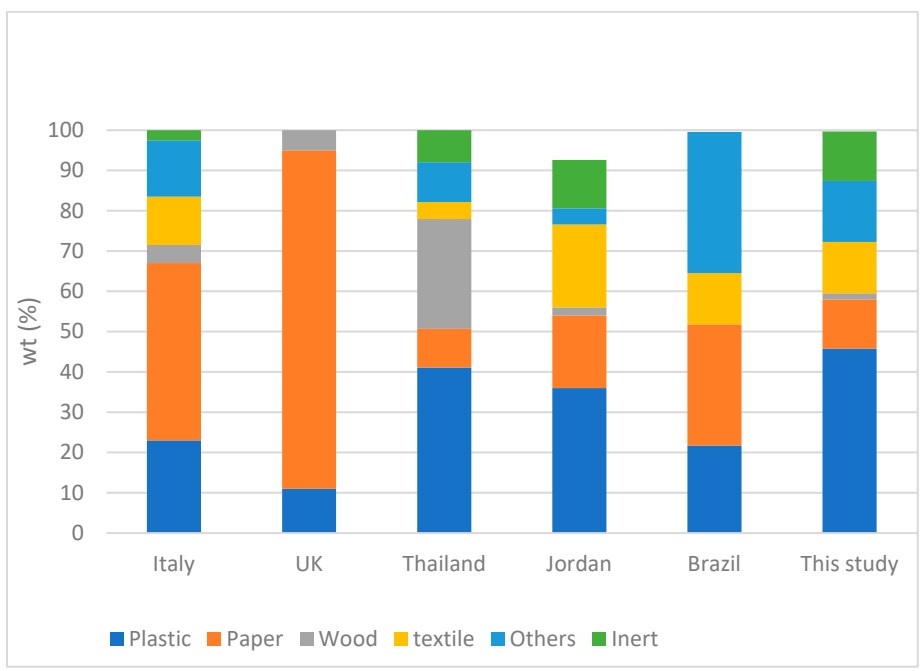

**Figure 6.** Average composition per unit of RDF from other studies [23,24,27,31].

### 3.3. Chemical Analysis

The analysis was carried out to ascertain the properties of the materials as RDF; proximate and calorific values are presented in Table 1. The content of ash in the proposed fuel ranged between 6.30% and 17.40%. However, the higher ash content observed was from the samples with organic fractions in option 2. The DM varied, ranging from 63% to 91%; the higher the moisture content, the smaller the combustible portion by weight per unit of RDF. The heating value obtained for option 1 RDF (28.66–30.24 MJ·kg) was higher than in option 2 (LHV; 19.73–23.75 MJ/kg), which confirms RDF improvement over MSW and the combustible portions as viable for RDF. The highest LHV was observed in samples with high plastic content. This is also higher than the mass MSW heating values observed in previous studies (Table 2) at 14.8–16.6 [50] and 18.4–22.9 MJ/kg [51] in Ghana, comparable to that of coal, 18–36 MJ·kg/4500–8500 kcal/kg [34] and higher than that of RDF samples from other studies in Figure 6. However, it is possible to decrease the moisture

in the RDF through pre-treatment such as bio-drying in an integrated system to increase the heating value. Significantly, the value of heat energy increased without organics/food waste. Therefore, the combustible fractions of the MSW reject fraction represent viable components for RDF, which meet LHV recommendations for industrial use [33].

**Table 1.** Proximate, heating values and elemental analysis of RDF samples.

| Parameter | T1 | T2 | T3 | Average | T4 | T5 | T6 | Average |
|---|---|---|---|---|---|---|---|---|
| Moisture (%) | 13.27 | 15.56 | 8.11 | 12.31 | 36.15 | 26.16 | 48.53 | 36.95 |
| DM (%TS) | 86.73 | 84.44 | 91.89 | 87.69 | 63.85 | 73.84 | 51.47 | 63.05 |
| Ash (%, db) | 6.30 | 7.10 | 6.60 | 6.67 | 17.40 | 16.40 | 11.89 | 15.23 |
| HHV (MJ/kg) | 31.59 | 30.55 | 32.09 | 31.41 | 21.94 | 25.17 | 21.09 | 22.73 |
| LHV (MJ/kg) | 30.24 | 28.66 | 29.57 | 29.49 | 20.77 | 23.75 | 19.73 | 21.42 |
| C (%) | 66.20 | 64.00 | 60.90 | 63.70 | 47.30 | 57.50 | 49.70 | 51.50 |
| H (%) | 9.80 | 9.20 | 9.00 | 9.33 | 5.70 | 6.90 | 6.60 | 6.40 |
| N (%) | 0.34 | 0.65 | 0.88 | 0.62 | 1.00 | 0.42 | 0.82 | 0.75 |
| S (%) | 0.10 | 0.11 | 0.13 | 0.11 | 0.14 | 0.09 | 0.19 | 0.14 |
| Cl (%) | 0.75 | 0.77 | 0.20 | 0.57 | 1.40 | 0.24 | 1.30 | 0.98 |
| Hg (mg/kg) | 0.06 | <0.05 | 0.08 | 0.07 | 0.16 | 0.07 | <0.05 | 0.076 |

S: sample, DM: dry matter, HHV: higher heating value, LHV: lower heating value.

The elemental analysis presented in Table 1 shows generally high carbon (47.30–66.20%) and hydrogen content. This shows a good energy potential, which translates into the heating values recorded. The nitrogen content was higher in samples of higher organic content than the others, which also recorded a higher NOx formation index. The chlorine content recorded from the experiments was 0.2–1.4%, which was within acceptable ranges for co-combustion and use in cement kilns. During combustion, a high concentration of chlorine enhances the formation of low melting point eutectics in fly ash, which condenses to induce corrosion. The sulfur content recorded was 0.09–0.19%. The presence of other alkali metals such as potassium and sodium stimulate high temperature corrosion, slagging, fouling and ash deposition, while in flue gases heavy metals (such as Zn, Pb) chlorides are present [52]. Some sources of chloride in MSW are reported from materials such as polystyrene, PVC and synthetic rubber [53]. On the other hand, kitchen waste also contributes to this contamination as a result of inorganic chlorides (NaCl, MgCl and KCl) present in various commodities used [53]. Source separation especially with kitchen waste from others can reduce the inorganic chloride content in the MSW. As a result, chlorine and sulfur are suitable in MSW fuel only in very low quantities, as shown in Table 3.

**Table 2.** Comparison of fuel from other sources.

| | Moisture (%) | Ash (%) | LHV (MJ·kg) | S (%) | Cl (%) | Ref |
|---|---|---|---|---|---|---|
| This study | 8.11–13.27 | 6.30–7.10 | 28.66–29.49 | 0.10–0.11 | 0.2–0.77 | |
| Coal | 1.8 | | 18–35.3 | 0.2–11 | | [5] |
| MSW—Ghana | 25–62 | 10.85–19.02 | 18.4–22.9 MJ/kg | - | - | [51] |
| Biomass—Ghana | 6.67–30 | 4–17 | 15.32–19.21 MJ/kg | 0.12–0.24 | - | [54,55] |
| RDF—Jordan | - | 16.7–19.6 | 15.85–16.70 | - | 0.95–1.03 | [31] |
| RDF—Thailand | 6.2–11.5 | 11.8–15.1 | 20.80–29.5 | 0.17–0.20 | 0.58–2.46 | [27] |
| RDF—India | 4.98–5.33 | 2.8–9.2 | 18.6–23.9 | 0.27–0.71 | 0.339–0.521 | [56] |

The heavy metals and trace elements for the useability index are present in Table 3. These concern industrial use such as fouling, slagging and corrosion at high temperatures. Mercury (Hg) values ranged from 0.06 to 0.16 mg/kg. There is also low to moderate,

high-temperature corrosion risk and heavy metal pollution potential recorded in terms of thermochemical applications, although they were within the acceptable range of assessment criteria. The RDF studied also already meets the minimum calorific value for RDF use as co–fuel in the cement industry, 4780 kcal/kg [57]. Table 2 also compares the analysis and energy content of RDF from MSW studied to other fuels derived from other waste and coal. The wide range of properties reported in this study and the literature may be explained on the basis of origin, heterogeneity and different composition of the MSW.

**Table 3.** Heavy metals and some useability indices.

| Property/Indices | T1 | T2 | T3 | T4 | T5 | T6 | Limits | Ref |
|---|---|---|---|---|---|---|---|---|
| Cd (mg/MJ) | 0.05 | 0.13 | 0.01 | 0.02 | 0.02 | 0.02 | ≤0.1–1.0 | [58] |
| HM (mg/MJ) | 11.79 | 26.57 | 7.63 | 18.84 | 8.86 | 15.03 | ≤15–30 | [58] |
| NOx emission (N%) | 0.34 | 0.65 | 0.88 | 1.00 | 0.42 | 0.82 | ≤0.5–1 | [2] |
| HTCR, (2S/Cl) | 0.27 | 0.29 | 1.30 | 0.20 | 0.75 | 0.29 | ≤0.2–0.5 | [59] |

HM: heavy metal pollution (sum of Sb, As, Pb, Cr, Co, Cu, Mn, Ni, V), HTCR: high-temperature corrosion risk.

*3.4. RDF Classification*

To use RDF for industrial purposes, it must conform to standard values associated with its use. Since there are no specific standards prescribed in Ghana or other developing countries, the characteristics and the classification from this study in Table 4 were compared with European standards in Table 5. The standards are classified as NCV since the fuel value translates into the economic aspects, the Cl which assesses the technological constraints and the Hg content and other heavy metals for the environmental impact [58]. The mineral content must also be sufficiently stable for optimal utilization. Thus, the characteristics of the RDF should enable storage and safe handling and satisfy environmental specifications. According to the results presented, the RDF obtained in the study was average in the category and classified as NCV 1, Cl 2 and Hg 3. All indices have allowable values in accordance with the European Norm EN 15359:2011 and that of utilization in a cement kiln set by EURITS [2]. Nevertheless, the RDF could be improved by pre-treatment and thermochemical processes to obtain a higher class for heavy metals.

**Table 4.** Classification of RDF from this study according to EN 15359:2011.

| Property | Statistical Measure | 1 | 2 | 3 | 4 | 5 | This Study |
|---|---|---|---|---|---|---|---|
| NCV (MJ/kg) | Average | ≥**25** | ≥20 | ≥15 | ≥10 | ≥3 | 29.49 |
| Cl (%) | Average | ≤0.2 | ≤**0.6** | ≤1.0 | ≤1.5 | ≤3.0 | 0.57 |
| Hg (mg/MJ) | Median | ≤0.02 | ≤0.03 | ≤**0.08** | ≤0.15 | ≤0.50 | 0.07 |
|  | 80th percentile | ≤0.04 | ≤0.06 | ≤**0.16** | ≤0.30 | ≤1.00 | 0.076 |

**Table 5.** Comparison of RDF characteristics with other quality standards.

| | **General** | | | **Co-Incineration in Cement Kiln** | | |
|---|---|---|---|---|---|---|
| Parameter | Italy | UK | Finland | EURITS | Switzerland | This study |
| LHV(MJ/kg) | >15 | >18.7 | 13–16 | >15 | 25.1–31.4 | 28.66–30.24 |
| Ash (%) | 20 | 12 | 5–10 | 5 | 0.6–0.8 | 6.30–7.10 |
| Moisture (%) | <25 | 7–28 | 25–35 | - | <10 | 8.11–15.56 |
| S (%) | | | <0.2–0.5 | 0.4 | <0.5 | 0.1–0.13 |
| Cl (%) | 0.9 | 0.3–1.2 | <0.15–1.5 | 0.5 | <1 | 0.20–77 |
| Hg(mg/kg) | | | <0.1–0.5 | | 0.02 | 0.05–0.08 |

Source [2], EURITS: European Association of Waste Thermal Treatment Companies.

Generally, the results obtained for the RDF samples showed high calorific value, low moisture and ash content and satisfactory chlorine content comparable to others shown in Table 5, meaning that the RDF is acceptable as an alternative fuel. Considering the characteristics of RDF obtained, it is concluded that the properties depend on the efficiency of the sorting system as well as the composition of the waste. In comparison, the improvement in fuel properties from MSW as RDF is comparable to other fuel sources utilized in Table 2. The overall findings of the characterization and experiments demonstrate that RDF from residual MSW in Ghana is a viable option and possible for integrated management plans.

### 3.5. Potential Energy Supply

These characteristics indicated that the RDF from the MSW reject fraction is a potential commercial-scale product and may be used as a substitute for fossil fuels. The market for RDF and its use in Ghana is developing. Co-combustion and co-processing represent attractive options for RDF utilization due to their high efficiency and sustainability [8]. Using the quantity of MSW and RDF fractions, a theoretical estimation of RDF and coal substitution is presented in Table 6. About 12 and 79 kilotons of RDF can be generated annually from the current MRFs in Accra and Kumasi, respectively, representing 5–25% of coal substitution. This is an advantage over the investment in the collection and transport of MSW for disposal without value addition, which, for example, is reported to cost USD 3.45 million annually in urban Ghana [60].

**Table 6.** Potential RDF Substitution.

| | Potential RDF (Kilotons/Year) | Energy (GJ/Year) | Energy (GWh/Year) | Coal Saving (Kilotons/Year) | Cost Saving (Million USD/Year) | $CO_2$ Saving (Kilotons/Year) | % of RDF to Coal Saving | Cost of RDF (Million USD/Year) | Net Savings (Million USD/Year) |
|---|---|---|---|---|---|---|---|---|---|
| F1 | 12.2 | 367.4 | 0.10 | 8.50 | 1.44 | 24.22 | 5.12 | 0.29 | 1.15 |
| F2 | 22.4 | 661.1 | 0.18 | 15.63 | 2.66 | 44.56 | 9.42 | 0.54 | 2.12 |
| F3 | 57.3 | 1643.6 | 0.46 | 40.10 | 6.82 | 114.29 | 24.16 | 1.39 | 5.43 |
| Total | 91.85 | 2672.08 | 0.74 | 64.23 | 10.92 | 183.07 | 38.69 | 2.23 | 8.69 |

The potential of RDF from MSW exists as an alternative measure for the landfill of non-degradable waste in Ghana. However, few efforts are in place to transform MSW into energy, though a few notable WtE projects have attempted to enter the energy mix. This is unlike advanced economies, where the market of RDF is robust and expanding, with a substitution rate of between 30–60% recorded in industries [8]. Opportunities and strengths can be harnessed to improve the WtE sector as well as weaknesses that need redress. Analysis based on the local setting is outlined for consideration. Table 7 presents findings of analysis from the WtE perspective. Locally, knowledge of the drivers of and barriers to integration of RDF from MSW for industrial use is essential to address measures that enable implementation. This effort will not only provide alternative fuel but also contribute to solving the MSW management challenges of cities in developing countries, such as Ghana.

**Table 7.** Analysis of barriers and drivers for alternative fuel from MSW.

| Drivers | Barriers |
|---|---|
| <ul><li>Resource availability: there is an increasing trend of MSW generation from fast urbanization, coupled with properties that meet energy recovery purposes.</li><li>Existence of potential industries to feed for co-combustion and electricity generation, etc.</li><li>Knowledge of gains: environmental, economic and social.</li><li>Creating synergy in cooperative projects towards a circular economy.</li><li>Alternative fuel from waste forms part of sustainable waste management in an integrated system.</li><li>Reduced GHG emissions (e.g., methane, $CO_2$) from waste disposal/landfilling and reliance on fossil fuels.</li><li>Industrial energy supply, e.g., as fuel for industrial use to support industrialization.</li><li>Diversity in the energy mix for sustainable production.</li><li>Possibility of promoting green solutions through business partnerships.</li></ul> | <ul><li>Regulations, lack of appropriate regulation and policies for the implementation of WtE, landfilling and waste by-product management practices.</li><li>Institutions: limited useful and reliable information on the WtE demand and supply.</li><li>Stakeholder networks and structural weaknesses.</li><li>Technical: developing the use of waste as an alternative fuel</li><li>Limited financial investments to implement and sustain initiatives.</li><li>Competition with power suppliers from conventional sources to industries.</li><li>Lack of clear roadmap from government and institutions on WtE implementation.</li><li>Lack of technical standards to guide implementation.</li><li>Funding</li><li>inadequate funding schemes for systems and infrastructure.</li><li>Low support from the local government; decentralization.</li></ul> |

## 4. Conclusions

MSW management remains a huge challenge in Ghana. Although efforts such as composting and recycling have emerged in recent years, landfilling without energy recovery, open dumping and indiscriminate disposal are prevalent. This study assessed RDF potential considering the existing material recovery facilities in Accra and Kumasi, the two largest cities of Ghana. The RF was made up of combustible materials including paper, plastics, textiles and wood, which are valuable components for RDF and can reduce landfill disposal. The study shows residual municipal solid waste is a good option for energy-efficient RDF. The RDF obtained in this study was classified as NCV 1, Cl 2 and Hg 2 under the EN 1539:2011 standard. Subsequently, it is estimated that RDF quantities can replace up to 25% of coal fuel utilized. The RDF shows higher LHV, lower moisture and lower ash, Cl, S and N than mass MSW. The findings show good potential for industrial fuel and the possibility of substituting for conventional fossil fuel use in Ghana. Diverting landfill waste disposal for refuse-derived fuel will also consequently protect soil/groundwater from contamination, microplastic contamination of water bodies and air pollution. Scalable systems and collaborations at small to medium scales will facilitate initiatives for the development of waste-to-energy. On the other hand, efforts and regulations from the government are necessary to implement such synergy towards the industrial use of alternative fuels such as RDF. Further studies with a focus on industrial symbiosis as well as sustainability assessment may provide the basis for collaborations and resource exchange between various industries. Research into the favorable conditions and determinants of symbiosis implementation is also of importance. Furthermore, studies towards pretreatment methods' suitability to improve the overall fuel properties will enhance the development of the RDF concepts. This study provides valuable insights into RDF production as a sustainable component of an integrated MSW management system, especially for developing countries, towards achieving SGDs and a circular economy.

**Author Contributions:** Conceptualization, K.S., S.N., G.B. and E.A.; Methodology, K.S. and U.B.; Formal Analysis, K.S.; Resources, K.S., S.N., G.B., E.A., U.B., M.H., N.S.A.D., E.A.A. and M.N.; Data Curation, K.S., S.N., G.B., E.A., U.B., M.H., N.S.A.D., E.A.A. and M.N.; Writing—Original Draft Preparation, K.S.; Writing—Review and Editing, K.S., S.N., G.B., E.A., U.B., M.H., N.S.A.D., E.A.A. and M.N.; Visualization, K.S.; Supervision, S.N., G.B. and M.N. All authors have read and agreed to the published version of the manuscript.

**Funding:** This work was supported under the project "Waste to Energy: Hybrid Energy from Waste as Sustainable Solution for Ghana", funded by the German Federal Ministry for Education and Research (BMBF), 03SF0591E, and the Open Access Publication Fund of the University of Rostock.

**Data Availability Statement:** The authors confirm that the data supporting the findings of this study are available within the article.

**Acknowledgments:** The authors would like to thank the facilities' management for their assistance in conducting the research as well as the anonymous reviewers, whose comments provided useful contributions to improving this manuscript.

**Conflicts of Interest:** The authors declare no conflict of interest.

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
