# Peer review of "Characterization of Municipal Solid Waste and Assessment of Its Potential for Refuse-Derived Fuel (RDF) Valorization"

_energies, doi:10.3390/en16010200_

Round 1
Reviewer 1 Report
please see the attachment

Reviewer 2 Report
Manuscript ID: energies-2100742
Type of manuscript: Article
Title: Characterization of Municipal Solid Waste and Assessment of its
Potential for Refuse Derived Fuel (RDF) Valorization
Authors: Khadija Sarquah *, Satyanarayana Narra, Gesa Beck, Uduak Bassey,
Edward Antwi, Michael Hartmann, Nana Sarfo Agyemang Derkyi, Edward A Awafo,
Michael NellesCharacterization of Municipal Solid Waste and Assessment of 2 its Potential for Refuse Derived Fuel (RDF) Valorization
Authors claimed that reuse and recycling are preferred strategies in waste management to keep waste resources higher up in the hierarchy and they mentioned that challenges are still pronounced in many developing countries, where disposal as a final solution is prevalent, particularly for municipal solid waste. On the other hand, refuse-derived fuel as energy recovery provides a sustainable option for managing mixed, 18 contaminated, and residual municipal solid waste (MSW). They also mentioned that this study provides one of the earliest attempts to assess refuse-derived fuel (RDF) from MSW in Ghana, a case study in the cities of Accra and Kumasi. They showed that the residual/reject fractions (RF) of MSW material recovery were characterized for thermochemical energy purposes. The studied materials had promising potential to be used as RDF. The combustible portions from the residual fractions formed good alternative fuel, RDF, under the 23 class I, II-III classification of the EN 15359:2011 standards.
They organized the manuscript very well and they used recent references. I am sure their study will contribute to future research in this area. To make the manuscript acceptable, the similarity is very high. Some paragraphs are copied and pasted and this is not acceptable.
Also, in the last paragraph of the introduction, please emphasize the novelty of those studies and why it is much different than previous studies.
Reviewer 3 Report
The paper presents the investigation on the fuel potential of MSW produces in Ghana together with its characteristics and assessment of possible utilization as an alternative fuel. The Authors conducted a very comprehensive analysis of the problem. The paper is in line with the scope of the Energy journal. This topic is of high importance and the scientific level of this paper is high thus it is attractive to the readers.
In general, the paper is well-written and has a good structure. Nevertheless, I have some comments:
1. Line 75: the waste-to-energy abbreviation is usually in capital letters (WtE).
2. Lines 115, 124, 132, 211, etc. look like editing errors
3. Figure 2 is of a bad quality
4. The problem of chlorine concentration and corrosion should be discussed in more detail, as it is one of the main factors determining the combustion potential of MSW and RDF. The Authors may develop the discussion in lines 296-300 with the use of the following literature positions:
https://doi.org/10.1016/j.pecs.2019.100789
https://doi.org/10.1016/j.fuel.2022.124749
5. In the conclusions, please provide a brief overview of the differences between the characters of MSW from Ghana and from developed countries (e.g. EU).
